# The Ladybird Beetles (Coleoptera: Coccinellidae) of La Palma

**DOI:** 10.3390/insects14070655

**Published:** 2023-07-22

**Authors:** Jerzy Romanowski, Piotr Ceryngier, Jaroslav Vĕtrovec, Karol Szawaryn

**Affiliations:** 1Institute of Biological Sciences, Cardinal Stefan Wyszyński University, Wóycickiego 1/3, 01-938 Warsaw, Poland; p.ceryngier@uksw.edu.pl; 2Buzulucká 1105, 50-003 Hradec Králové, Czech Republic; jerryvetrak@seznam.cz; 3Museum and Institute of Zoology, Polish Academy of Sciences, Wilcza 64, 00-679 Warsaw, Poland; k.szawaryn@gmail.com

**Keywords:** biodiversity, Canary Islands, alien species, new records, Coccinelloidea

## Abstract

**Simple Summary:**

The ladybird beetle fauna of the Canary Islands is quite specific due to the presence of a number of species that do not occur anywhere else (the so-called endemic species). However, many ladybirds recorded in the archipelago are relatively recent arrivals from various parts of the world, as shown by our previous surveys carried out in several of the Canary Islands, including Fuerteventura, Lanzarote, Gran Canaria, and El Hierro. In this paper, we analyze the ladybird fauna of La Palma, one of the western islands of the archipelago, based on our field survey and already published data. The survey resulted in the recording of 26 species, seven of which had not previously been recorded on La Palma, and two of these seven had not been recorded on any of the Canary Islands. Combining our data and literature reports gives a figure of at least 35 ladybird species recorded to date on La Palma. This is fewer than on the central islands of the Canary archipelago (Gran Canaria and Tenerife), but more than on the other four islands (Fuerteventura, Lanzarote, La Gomera, and El Hierro). This study confirms previous observations that the Canary Islands are often colonized by exotic ladybird species.

**Abstract:**

This paper provides new data on the ladybird beetles (Coccinellidae) of La Palma, one of the western islands of the Canarian archipelago. The field survey of 54 study sites resulted in recording 2494 ladybird individuals belonging to 26 species. Seven of the species recorded were new to La Palma, including two, *Harmonia quadripunctata* (Pontoppidan) and *Nephus reunioni* (Fürsch), which were not registered so far on any of the Canary Islands. *Novius conicollis* (Korschefsky) is synonymized with *N. cruentatus* (Mulsant). Taking our survey and literature reports into account, a total of at least 35 species of Coccinellidae have so far been recorded on La Palma. This richness in species is lower compared to that of the central islands of the Canarian archipelago, Gran Canaria (42 species) and Tenerife (41 species), but higher than that of the remaining four islands (between 22 and 27 species). The detection of two alien species new to La Palma, *Nephaspis bicolor* Gordon and *Nephus reunioni* (Fürsch), confirms earlier observations that colonization of the Canary Islands by ladybird species of exotic origins seems to be a frequent phenomenon.

## 1. Introduction

There are over 6000 species of Coccinellidae globally of which about 55 species were reported to occur on the Canary Islands [1,2,3,4]. A significant proportion of them are Canarian or Macaronesian endemics, but many species currently occurring on the Canaries are non-indigenous to this archipelago [3,5]. Some, such as *Olla v-nigrum* (Mulsant), *Nephaspis bicolor* Gordon, *Pharoscymnus flexibilis* (Mulsant), and *Cheilomenes propinqua* (Mulsant), have become established there only in recent years [2,3,4,6], indicating a high rate of colonization and a need to monitor the changes in the Canarian ladybird fauna.

Ladybirds found on the Canary Islands, both indigenous and non-indigenous, are almost exclusively predatory (only one mycophagous species, *Vibidia duodecimguttata* (Poda), was found by Uyttenboogaard [7] on Gran Canaria). No members of the herbivorous tribe Epilachnini have so far been recorded there, although in nearby Northwest Africa and Southwest Europe, several species of this tribe are found [8,9,10,11]. This discrepancy is consistent with Becker’s [12] finding that the ratio of predatory to herbivorous Coleoptera tends to be higher on oceanic islands than in the nearest mainland areas. According to Becker [12], the likely explanation for this is that predators may find it easier to establish and persist on islands than herbivores because the former are often food generalists in contrast to more food-specialized, host plant-dependent herbivores. It is also well known that generalist predators establishing themselves on islands may have especially destructive effects on those islands’ native biota [13].

This study is a continuation of our faunistic research on the Coccinellidae in the Canary Islands. Our recent surveys on the islands of Fuerteventura [6,14], Lanzarote [15], Gran Canaria [4], and El Hierro [16] revealed many ladybird species not previously recorded on those islands, including several alien species.

## 2. Materials and Methods

The Canary Islands lie in the northeast Atlantic Ocean near the African coast and are part of the Mediterranean Basin biodiversity hotspot [17]. La Palma is the northwesternmost island, and the most distant from the African mainland. This volcanic island has a relatively small area (706 km^2^) and reaches a maximum elevation of 2426 m.a.s.l. at Roque de los Muchachos. The mild subtropical Atlantic climate of the island is strongly influenced by humid trade winds that transport moisture from the northeast. As a result, the northern to eastern areas experience relatively stable humid conditions, while the western to southern regions are characterized by low and infrequent precipitation [18]. The vegetation of La Palma includes a wide range of habitats, such as laurisilva and *Pinus canariensis* forests, coastal habitats, and various kinds of scrub vegetation. In farmland areas, plants of agricultural interest are abundant, and in anthropogenic habitats, decorative plants sustained with irrigation are cultivated (Figure 1).

Ladybirds were recorded at 54 sites on La Palma (Table 1) using standard collecting methods, such as a beating tray, a sweeping net, or through direct observation. Although some of the caught ladybird individuals were released after their identification, each individual was noted. The majority of specimens were recorded by J. Romanowski and P. Ceryngier between 16 and 21 June 2021. Some material collected earlier, in 2013, 2014, 2018, and 2019, by A. Machado, T. Staněk, and J. Krátký was also used in this study. The voucher specimens are stored in the insect collection in the Institute of Biological Sciences, Cardinal Stefan Wyszyński University in Warsaw, and in the private collections of Jaroslav Větrovec. Species identification was based on morphological and anatomical features, including the form of reproductive organs (see [4,5,14,16]), and individuals collected in the larval and pupae stages were reared into adults in the laboratory. Habitus images were taken using a Leica MZ 16 stereo microscope with an IC 3D digital camera attached. The photographs of the genitalia were taken using an Olympus DP23 digital camera attached to an Olympus BX43F compound microscope. Final images were produced using Helicon Focus 5.0 × 64 and Adobe Photoshop CS6 software. The systematic arrangement of Coccinellidae used in this study follows Che et al. [19].

## 3. Results

Altogether, 2494 ladybird individuals (2390 adults, 88 larvae, and 16 pupae) belonging to 26 species (including one species identified to the genus level) were recorded in this study. Detailed data on all the recorded species are provided below. Morphological and anatomical details of several species of special interest are photographed.

Microweiseinae Leng, 1920Serangiini Pope, 1962


***Delphastus catalinae* (Horn, 1895)**


**Material examined:** From Cubo de la Galga, Fuente del Toro, Las Caletas, Mazo, Santa Cruz, and Tazacorte: 17.VI–21.VI.2021, a total of 94 specimens were collected mostly from *Ficus* sp., *Nerium oleander* L., *Prunus dulcis* (Mill.) D.A. Webb, and *Punica granatum* L.

**Distribution:** This species has been widely used as a biocontrol agent in various parts of the world. It is thought to occur natively in Colombia, Mexico, southern California (USA), and on the island of Trinidad [20]. In the Canary Islands, it was previously recorded on La Palma, La Gomera, Tenerife, Gran Canaria, Fuerteventura, and Lanzarote [4].

Coccinellinae Latreille, 1807Stethorini Dobzhansky, 1924


***Stethorus tenerifensis* Fürsch, 1987**


**Material examined:** From Fuencaliente, Fuente del Toro, Las Caletas, Mazo, Malpaíses, Pino de la Virgen, Santa Cruz, and Tijarafe: 17.VI–20.VI.2021, a total of 15 exx. were collected from various plants including *P. dulcis*, *N. oleander*, *Ficus* sp., and *Hedera* sp.

**Distribution:** This is an endemic Macaronesian species, reported to be on all main islands of the Canary Islands [1,15] as well as Madeira [11].


***Stethorus wollastoni* Kapur, 1948**


**Material examined:** From Llano Negro: 19.III.2018, a total of three exx. from *Genista canariensis* L. and *Ilex canariensis* Poir. were collected; from Fuente del Toro: 18.06.2021, one male was collected.

**Distribution:** This is a Macaronesian species reported on all the Canary Islands [1] and Madeira [11].

Coccinellini Latreille, 1807


***Adalia bipunctata* (Linnaeus, 1758)**


**Material examined:** From Fuente del Toro: 18.VI.2021, 10 exx. from *P. dulcis*, were collected; from Tijarafe: 18.VI.2021, five exx. from *P. dulcis* were collected; from El Paso: 20.VI.2021, one ex. was collected from *Ficus* sp.; from Mazo: 20.VI–21.VI.2021, two exx. (one adult, one pupa) from *Ficus* sp. were collected.

**Distribution:** This species is widespread in the Holarctic, and was also introduced to Australia, New Zealand, Africa, and South America [21,22,23]. In the Canary Islands, it has been found on La Palma, La Gomera, and Tenerife [1].

**Remarks:** Specimens of *A. bipunctata* collected in this study represent several color forms, including f. *typica*, *sexpustulata*, *quadrimaculata*, and *fasciatopunctata*. The lattermost is mainly known from the Middle East and central Asia [24,25]. Its presence on La Palma may, therefore, indicate the source of the Canarian population of *A. bipunctata.*


***Coccinella miranda* Wollaston, 1864**


**Material examined:** From Los Braseros: 26.I.2013, 12 exx. were collected; from Barranco de las Angustias, Caldera de Taburiente, Las Manchas, Mirador de Mendo, Montaña de Tagoja, Montes de Luna, and Valencia: 13.III–19.III.2018, a total of 127 exx. were collected; from Barranco del Carmen Dorador, El Pilar, Fuencaliente, Mazo, Mirador Llano de las Ventas, Pared Vieja, Santa Cruz, Tijarafe, La Rosa, and Fuente Olén: 16.VI–21.VI.2021, a total of 169 exx. (167 adults, one pupa, one larva) were collected from *Pinus canariensis,* Fabaceae, and herbaceous plants.

**Distribution:** This is an endemic Canarian species reported to be on all islands of the archipelago except Lanzarote [1,15].


***Coccinella septempunctata algerica* Kovář, 1977**


**Material examined:** From Pared Vieja: 16.VI.2021, two exx. were collected; from Malpaíses: 17.VI.2021, one ex. was collected; from the LP-4 roadside: 18.VI.2021, one ex. from *Pinus canariensis* was collected.

**Distribution:** This subspecies occurs mainly in North Africa. It is reported to be on all the Canary Islands [1].


***Harmonia quadripunctata* (Pontoppidan, 1763)**


**Material examined:** From Barranco del Carmen Dorador: 18.VI.2021, one ex. was collected, and 19.VI.2021, three exx. were collected, all from *Pinus canariensis.*

**Distribution:** This is a Palaearctic species [23], not previously recorded on the Canary Islands.


***Hippodamia variegata* (Goeze, 1777)**


**Material examined:** From El Pilar, Las Indias, Malpaíses, Mazo, Pared Vieja, Santa Cruz, and Tazacorte: 16.VI–21.VI.2021, a total of 38 specimens (26 adults, 12 larvae) were collected from *N. oleander*, *Tamarix* sp., *Euphorbia* sp., ferns, and herbaceous flowering plants.

**Distribution:** This species is widely distributed in the Palaearctic [23] and in many regions outside of its native range, such as South and North America, Africa, India, Australia, New Zealand, and even the remote Easter Island in the Pacific [26,27,28,29,30]. It is common on all islands of the Canary archipelago [1].


***Myrrha octodecimguttata* (Linnaeus, 1758)**


**Material examined:** From El Pilar: 16.VI.2021, seven exx. were collected, and 21.VI.2021, eight exx. were collected, all from *Pinus canariensis.*

**Distribution:** This is a Palaearctic species [23], recorded in the Canary archipelago on La Gomera [1], and recently also on Gran Canaria [4] and El Hierro [16]. It is new to La Palma.

Novini Mulsant, 1846


***Novius cardinalis* (Mulsant, 1850)**


**Material examined:** From Breña Baja: 25.I.2013, one ex. was collected; from Barlovento, Barranco de los Hombres, and Las Caletas: 16.III-17.III.2018, total of three specimens from *Euphorbia* sp., *Kleinia* sp., and *Vicia* sp. were collected*;* from Buenavista de Arriba, Cudad Alta, Cubo de la Galga, Los Llanos, Mazo, and Santa Cruz: 18.VI–21.VI.2021, a total of 31 specimens (24 adults, three larvae, four pupae) were collected from *N. oleander*, *Hibiscus* sp., *Phoenix canariensis*, and palms.

**Distribution:** This species is native to Australia but introduced in many regions throughout the world [31]. It is present on all the Canary Islands [1,14,15].


***Novius cruentatus* (Mulsant, 1846)**


*Novius conicollis* Korschefsky, 1935: 1, **syn. nov.** (Figure 2A–I and Figure 3A).

**Material examined:** From Montaña de Tagoja: 15.III.2018, one ex. was collected; from Barranco del Carmen Dorador: 18.VI.2021, 28 exx. were collected (25 adults, three larvae), and 19.VI.2021, 24 exx. were collected, all from *Pinus canariensis.*

**Distribution:** This is a species mainly found in central and southern Europe [23]. It is reported on the Canary Islands either as *N. cruentatus* or *N. conicollis*, on El Hierro [1,32], La Palma [1,33,34], Tenerife [1,35], and Gran Canaria [1,4].

**Remarks:** *N. cruentatus* is a specialized predator of *Palaeococcus fuscipennis* (Burmeister), a giant scale (Hemiptera: Monophlebidae) associated with several pine species [36,37,38]. In the Canary Islands, *N. cruentatus* is usually found on *Pinus canariensis* ([4,34], this study). Korschefsky [33] considered the specimens collected on La Palma as representing a separate species, different from *N. cruentatus* and placed them in a newly described species, *N. conicollis*. However, examination of male genitalia in specimens collected in this study revealed no difference between them and the European specimens of *N. cruentatus*, and therefore we here synonymize *Novius conicollis* Korschefsky, 1935 with *Novius cruentatus* (Mulsant, 1846).


***Novius canariensis* Korschefsky, 1935**


**Material examined:** From Las Caletas II: 17.III.2018, one ex. was collected (adult female); from Puerto de Puntagorda: 18.III.2018, one ex. was collected.

**Distribution:** This is an endemic Canarian species, reported to be on Gran Canaria [33], Tenerife [35], and El Hierro [16]. It is new to La Palma.

Scymnini Mulsant, 1846

***Nephaspis bicolor* Gordon, 1982** (Figure 3B and Figure 4A–L)

**Material examined:** From Santa Cruz: 21.VI.2021, 10 exx. were collected from a palm tree heavily infested with coccids and aleyrodids.

**Distribution:** This species is native to Trinidad and was introduced into Hawaii [39] and the Canary Islands [2]. Between 2005 and 2010, *N. bicolor* was introduced on three islands of the archipelago, El Hierro, La Palma, and Tenerife, and in 2015 it was found to be well established on Tenerife [2]. Our present record indicates that this species has also been established on La Palma.

***Nephus*** **(*Nephus*) *flavopictus* (Wollaston, 1854)**

**Material examined:** From Las Caletas II: 28.I.2013, one ex. was collected; from Cubo de la Galga: 8.II.2014, one ex. was collected; from Puerto Naos: 9.II.2014, two exx. were collected; from Montes de Luna: 13.III.2018, one ex. was collected; from Cueva de los Palmeros: 14.III.2018, one ex. was collected; from Barranco de los Hombres: 16.III.2018, three exx. were collected; from Puntagorda: 18.III.2018, one ex. was collected; from Playa de Nogales: 19.VI.2019, four exx. were collected; from Barranco del Carmen Dorador, Buenavista de Arriba, Fuencaliente, Fuente del Toro, Las Caletas, Malpaíses, Mazo, Montaña de Mago, Puerto Espíndola, San Andrés, Santa Cruz, Tazacorte, and Tijarafe: 17.VI–21.VI.2021, a total of 50 exx. were collected mostly from succulents, *Euphorbia* sp., *Ficus *sp.**, *N. oleander,* and *Hedera* sp.

**Distribution:** This is a Macaronesian species, reported on all the Canary Islands [1].

***Nephus*** **(*Nephus*) *incisus* (Har. Lindberg, 1950)**

**Material examined:** From Las Caletas: 17.VI.2021, one ex. was collected; from Mazo: 20.VI.2021, one ex. was collected.

**Distribution:** Until recently, this species was considered to be endemic to the Canary Islands, but new data showed its presence in continental Spain, Algeria, and Cape Verde [40]. It is found on all islands of the Canary archipelago [1,14,15,16].


***Nephus* (*Geminosipho*) *reunioni* (Fürsch, 1974)**


**Material examined:** From Fuente del Toro: 18.VI.2021, one ex. was collected (larva bred to adulthood) from *N. oleander*; from Los Llanos: 18.VI.2021, 36 exx. were collected; from Santa Cruz: 19.VI.2021, three exx. were collected from *P. granatum*, and 21.VI.2021, 40 exx. were collected mostly from *N. oleander*; from Mazo: 21.VI.2021, one ex. was collected from *Artemisia* sp.

**Distribution.** This species is endemic to the Mascarene Island of Réunion [41]. In Macaronesia, it was previously reported in the Azores and Madeira [11], but not on the Canary Islands.


***Scymnus* (*Mimopullus*) *cercyonides* Wollaston, 1864**


**Material examined:** From El Tablado: 27.I.2013, one ex. was collected, 13.II.2014, two exx. were collected, and 16.III.2018, two exx. were collected; from Puerto de Puntagorda: 18.III.2018, two exx. were collected; from Mazo: 16.VI.2021, one ex. was collected from *Euphorbia* sp.; from Las Caletas: 17.VI.2021, one ex. was collected from *N. oleander.*

**Distribution:** This is an endemic Canarian species, reported to be on all islands of the archipelago except the easternmost Fuerteventura and Lanzarote [1].


***Scymnus (Pullus) canariensis* Wollaston, 1864**


**Material examined**: From Los Braseros: 26.I.2013, two exx. were collected; from Montes de Luna: 13.III.2018, 20 exx. were collected; from La Fajana: 16.III.2018, 13 exx. were collected; from Las Caletas II: 17.III.2018, 12 exx. were collected; from Puntagorda: 18.III.2018, 29 exx. were collected; from Barranco del Carmen Dorador, Breña Alta, Buenavista de Arriba, Cudad Alta, Cubo de la Galga, El Granel, El Pilar, El Paso, Fagundo, Fuencaliente, Fuente del Toro, Garafía, Las Caletas, Las Indias, Los Llanos, Malpaíses, Mazo, Mirador de los Dragos, Mirador Llano de las Ventas, Mirador del Time, Montaña de Mago, Pared Vieja, Pico de la Nieve, Pino de la Virgen, Puerto Espíndola, San Andrés, San Juan, Santa Cruz, Tazacorte, and Tijarafe: 16.VI–21.VI.2021, a total of 1147 exx. were collected from various plants including *N. oleander, Prunus dulcis, Hibiscus* sp.*, Phoenix canariensis*, *Pinus canariensis*, *Euphorbia* sp., *Hedera* sp., *Olea europaea* L., *Arundo donax* L., and herbaceous vegetation.

**Distribution:** This is an endemic Canarian species, recorded throughout the Canary Islands [1].


***Scymnus* (*Scymnus*) *nubilus* Mulsant, 1850**


**Material examined:** From Fuente del Toro, Los Llanos, Mazo, Mirador del Time, Puerto Espíndola, San Andrés, Santa Cruz, and Tazacorte: 16.VI–21.VI.2021, a total of 48 exx. were collected mostly from *N. oleander*, *Ficus* sp., *A. donax*, Tamarix sp., *Ricinus communis* L., and *Phoenix canariensis*.

**Distribution:** This species is widely distributed in the Mediterranean Basin and the Middle East [23]. It is previously recorded on all islands of the Canary archipelago except La Palma [1,14,15].

Diomini Gordon, 1999

***Diomus* sp.** (Figure 3C and Figure 5A–D)

**Material examined:** From Santa Cruz: 21.VI.2021, one ex. was collected (female) from a palm tree.

**Remarks:** We are unable to determine the species affiliation of this single female. The genus is represented in the Canary archipelago by one species: *Diomus gillerforsi* Fürsch, recorded on La Palma and Tenerife [1,42].

Azyini Mulsant, 1850


***Cryptolaemus montrouzieri* Mulsant, 1853**


**Material examined:** From Cudad Alta, Fuencaliente, Fuente del Toro, Malpaíses, Mazo, Mirador de los Dragos, Santa Cruz, and Tijarafe: 17.VI–21.VI.2021, a total of 150 exx. were collected (112 adults, 10 larvae, 28 pupae) from *N. oleander*, *Dracaena* sp., *Opuntia ficus-indica* (L.) Mill., *P. dulcis*, *Phoenix canariensis*, and *Euphorbia* sp.

**Distribution:** This species is a well-known biocontrol agent of Australian origin, established in many warmer regions worldwide [43]. It has been recorded on all seven islands of the Canary archipelago [1,14,15].

Chilocorini Mulsant, 1846


***Chilocorus canariensis* Crotch, 1874**


**Material examined:** From Pico de la Cruz: 11.II.2014, one ex. was collected; from La Fajana: 16.III.2018, one ex. was collected; from Las Caletas II: 17.III.2018, two exx. were collected; from Las Caletas: 17.VI.2021, three exx. were collected from *Euphorbia* sp.; from San Andrés: 19.VI.2021, one ex. was collected; from Santa Cruz: 19.VI.2021, eight larvae were collected from palms.

**Distribution:** This species is endemic to the Canary Islands, reported to occur on all islands of the archipelago [1].


***Parexochomus nigripennis* (Erichson, 1843)**


**Material examined:** from Barranco del Carmen Dorador: 19.VI.2021, eight larvae were collected.

**Distribution:** This species occurs in the Mediterranean and Middle Eastern regions as well as in northwestern India, Pakistan, and the Afrotropical region [23,44]. It has been previously recorded on all islands of the Canary archipelago except La Palma [1].

Sticholotidini Pope, 1962


***Pharoscymnus decemplagiatus* (Wollaston, 1857)**


**Material examined:** From Las Caletas II: 12.II.2014, one ex. was collected; from Montes de Luna: 13.III.2018, five exx. were collected; from La Fajana: 16.III.2018, one ex. was collected; from Puntagorda: 18.III.2018, three exx. were collected from *Cistus* sp.; from Barranco del Carmen Dorador, Fagundo, Fuente del Toro, Las Caletas, Mazo, Mirador de los Dragos, Montaña de Mago, San Andrés, Santa Cruz, Tazacorte, and Tijarafe: 17.VI–20.VI.2021, a total of 151 specimens (124 adults and 27 larvae) were collected from *N. oleander*, *Tamarix* sp., *Pinus canariensis,* and *Dracaena* sp.

**Distribution:** This is a Macaronesian species, long known from Madeira [45] and the Canaries [46], and recently also found in the Azores [47]. In the Canary archipelago, it has been recorded on all islands [1,4,14,15].

Coccidulini Mulsant, 1846


***Rhyzobius litura* (Fabricius, 1787)**


**Material examined:** From Los Braseros: 26.I.2013, one ex. was collected; from El Tablado: 29.I.2013, three exx. were collected; from Puntagorda: 29.I.2013, two exx. were collected; from Montes de Luna: 30.I.2013, one ex. was collected; from San Juan de Puntallana: 8.II.2014, two exx. were collected; from Barranco de los Hombres: 13.II.2014, three exx. were collected; from Montaña de Tagoja: 15.III.2018, one ex. was collected; from Barlovento: 16.3.2018, five exx. were collected; from La Fajana: 16.III.2018, one ex. was collected; from Tijarafe: 18.III.2018, two exx. were collected; from El Granel: 19.VI.2021, one ex. was collected from *Rubus* sp.

**Distribution:** This species is widely distributed in Europe and North Africa and is also reported to occur in the Asiatic part of Turkey [23]. It is found on all the Canary Islands [1].


***Rhyzobius lophanthae* (Blaisdell, 1892)**


**Material examined:** From Barranco de los Hombres: 13.II.2014, one ex. was collected; from Puerto de Puntagorda: 18.III.2018, one ex. was collected; from Buenavista de Arriba, Cudad Alta, El Paso, Fuencaliente, Fuente del Toro, Las Caletas, Las Indias, Los Llanos, Malpaíses, Mazo, Mirador de los Dragos, San Andrés, Santa Cruz, and Tijarafe: 16.VI–21.VI.2021, a total of 82 exx. were collected mostly from *Phoenix canariensis*, *Dracaena* sp., Cycas sp., *Hibiscus* sp., *Ficus* sp., *Euphorbia* sp., and *Tamarix* sp.

**Distribution:** This is an Australian species, introduced in many regions of the world as a biocontrol agent [48]. It is found on all islands of the Canary archipelago [1,49].

## 4. Discussion

Prior to this study, 28 ladybird species were reported to occur on La Palma (Table 2). We failed to find 10 of them but found seven species not previously reported as occurring on this island, including two (*Harmonia quadripunctata* and *Nephus reunioni*) that are new to the whole Canary archipelago. Another species recorded by us, *Nephaspis bicolor*, was released several years ago on La Palma [2] but had not been found as established there prior to this study. We also collected a single female of the *Diomus* sp. whose species affiliation could not be determined, but whose spermatheca is different from that of the Mediterranean and Middle Eastern *D. rubidus* (Motschulsky). Perhaps this female belongs to *D. gillerforsi* Fürsch, a species described on the basis of one male collected on Tenerife [42]. To solve this problem, more material representing both sexes is needed.

Altogether, the literature reports and our data indicate the occurrence of 35–36 species of Coccinellidae on La Palma. This is fewer than those reported so far on Gran Canaria (42 species) and Tenerife (41 species) [4], but clearly more than on Fuerteventura (27 species), La Gomera (24 species) [1,49], Lanzarote (23 species) [15], and El Hierro (22 species) [16].

Of the species recorded in this study as new to La Palma, one, *N. canariensis*, is considered to be endemic to the Canary Islands [4]; four, *H. quadripunctata, M. octodecimguttata, S. nubilus,* and *P. nigripennis*, occur both on the Canary Islands and the neighboring regions of North Africa and Europe [23]; and two, *N. bicolor* and *N. reunioni*, are alien species, native to the remote islands of Trinidad (Lesser Antilles in the Atlantic Ocean) and Réunion (Mascarenes in the Indian Ocean), respectively [39,41].

The La Palma population of *N. bicolor* is almost certainly derived from individuals introduced to the Canary Islands for biocontrol of whiteflies (Hemiptera: Aleyrodidae). As reported by Rizza Hernández and Hernández Suárez [2], between 2005 and 2010, *N. bicolor* imported from Trinidad was released at several sites on the islands of La Palma, El Hierro, and Tenerife, and in 2015, abundant, established populations were recorded on Tenerife. On La Palma, a total of only 36 individuals were released in Puerto Naos on the western coast of the island in 2008 and 2009. After about 12 years, we found this ladybird in Santa Cruz, on the opposite (eastern) coast. The distance between these two localities is not great in a straight line (less than 20 km) but, as the island is baffled meridionally by a steep mountain chain, the direct spread of the beetle from west to east may have been impeded. We, therefore, suppose that *N. bicolor* has dispersed along the coastal areas of La Palma or arrived there from Tenerife.

*Nephus reunioni* was introduced in the 1970s and 1980s to some Mediterranean countries and the former USSR as a biocontrol agent against mealybugs (Hemiptera: Pseudococcidae), and subsequently established in Albania, France, Greece, Italy, Portugal, and Spain [61]. Later, its occurrence in the Azores and Madeira was documented [11,62].

In addition to the two species mentioned above, five other alien species have been recorded on La Palma. Four of them, American *Delphastus catalinae* and Australian *Novius cardinalis*, *Cryptolaemus montrouzieri,* and *Rhyzobius lophanthae* were found in this study, but we did not manage to collect the fifth species, American *Olla v-nigrum*. Most of these alien ladybirds are specialized predators of whiteflies (*N. bicolor*, *D. catalinae*) or scale insects (*N. reunioni*, *N. cardinalis*, *C. montrouzieri*, and *R. lophanthae*), but *O. v-nigrum* is a generalist predator, most often preying on aphids and psyllids, with a high potential invasiveness [5]. To date, it has become established on many oceanic islands and archipelagos in the Pacific and Indian Oceans (Easter Island, Hawaii, Midway, Guam, Tahiti, New Caledonia, Japan, and Réunion) [63,64,65], and has recently started to spread in Macaronesia. On the Canary Islands, *O. v-nigrum* has been observed since 2014, first on Tenerife and La Palma and then on Lanzarote and Gran Canaria [4,15,66]. In 2020, it was also recorded in Madeira [5].

Another generalist predator, the Asiatic *Harmonia axyridis* (Pallas), can also be expected to make an imminent appearance on La Palma and other islands in the Canary archipelago. Specimens of this well-known, now nearly cosmopolitan invasive species [67] were already recorded on Tenerife in 2003 and 2004 [49,68] and recently in 2022 [69]. *H. axyridis* could not establish, despite its deliberate introductions in the Azores between 1988 and 1995 [70], and the first findings of a reproducing population of *H. axyridis* in Macaronesia have been reported from Funchal (Madeira) in 2019 and 2020 [5,11].

## Figures and Tables

**Figure 1 insects-14-00655-f001:**
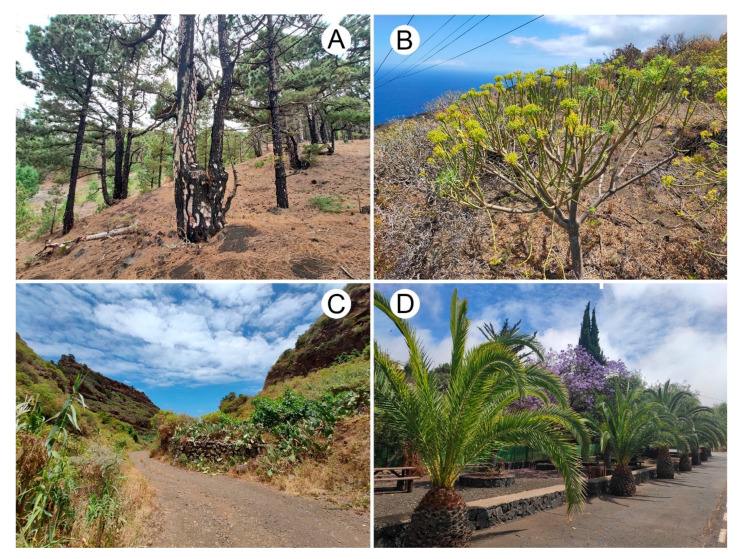
Some of the habitats surveyed in this study: (**A**) A pine forest with *Pinus canariensis*; (**B**) Scrub vegetation with *Euphorbia* spp.; (**C**) Vegetation with *Arundo donax* and *Opuntia ficus-indica* near the roadside; (**D**) Decorative vegetation.

**Figure 2 insects-14-00655-f002:**
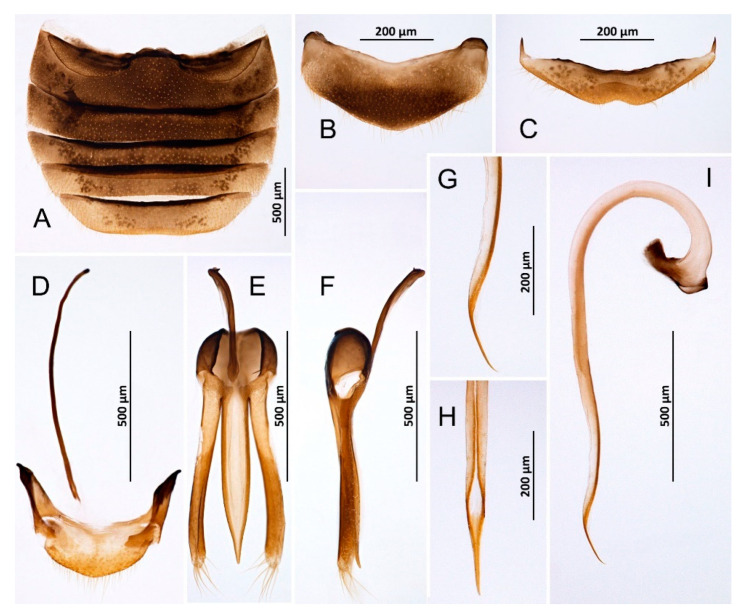
*Novius cruentatus,* male (**A**) Abdomen; (**B**) Abdominal tergite VIII; (**C**) Ventrite 6; (**D**) Abdominal segments IX and X; (**E**) Tegmen, inner; (**F**) Tegmen, lateral; (**G**) Tip of penis, lateral; (**H**) Tip of penis, inner; (**I**) Penis, lateral.

**Figure 3 insects-14-00655-f003:**
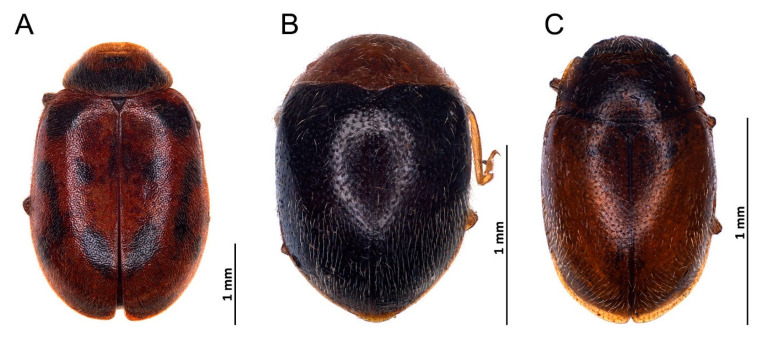
Habitus. (**A**) *Novius cruentatus*; (**B**) *Nephaspis bicolor*; (**C**) *Diomus* sp.

**Figure 4 insects-14-00655-f004:**
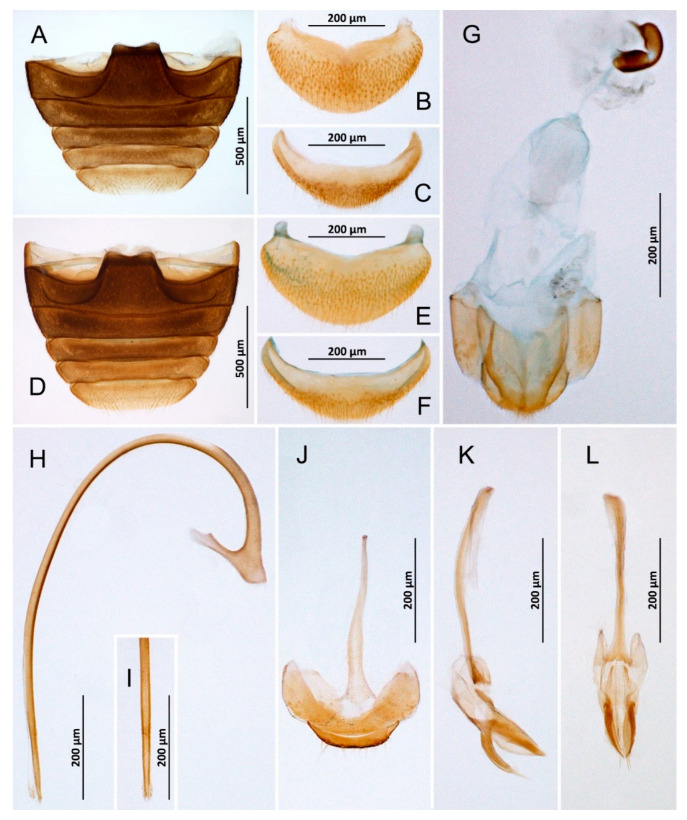
*Nephaspis bicolor.* (**A**) Abdomen, male; (**B**) Abdominal tergite VIII, male; (**C**) Ventrite 6, male; (**D**) Abdomen, female; (**E**) Tergite VIII, female; (**F**) Ventrite 6, female; (**G**) Female terminalia and genitalia; (**H**) Penis, lateral; (**I**) Tip of penis, inner; (**J**) Abdominal segments IX and X, male; (**K**) Tegmen, lateral; (**L**) Tegmen, inner.

**Figure 5 insects-14-00655-f005:**
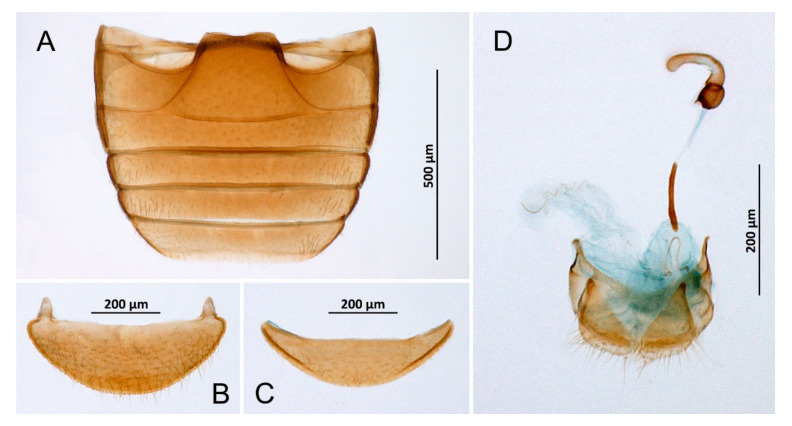
*Diomus* sp., female. (**A**) Abdomen; (**B**) Abdominal tergite VIII; (**C**) Ventrite 6; (**D**) Terminalia and genitalia.

**Table 1 insects-14-00655-t001:** Collection sites of ladybird beetles on La Palma.

Location	Coordinates
Barlovento	28°49′45″ N 17°47′23″ W
Barranco del Carmen Dorador	28°42′11″ N 17°46′43″ W
Barranco de las Angustias	28°38′35″ N 17°56′02″ W
Barranco de los Hombres	28°49′51″ N 17°52′05″ W
Breña Alta	28°38′52″ N 17°46′01″ W
Breña Baja	28°37′59″ N 17°46′35″ W
Buenavista de Arriba	28°41′11″ N 17°47′13″ W
Caldera de Taburiente	28°44′23″ N 17°49′38″ W
Cubo de la Galga	28°46′00″ N 17°46′11″ W
Cudad Alta	28°40′17″ N 17°47′51″ W
Cueva de los Palmeros	28°30′26″ N 17°52′23″ W
El Granel	28°45′15″ N 17°45′09″ W
El Paso	28°39′08″ N 17°51′26″ W
El Pilar	28°36′50″ N 17°50′10″ W
El Tablado	28°50′08″ N 17°52′34″ W
Fagundo	28°45′55″ N 17°58′33″ W
Fuencaliente	28°29′36″ N 17°50′41″ W
Fuente del Toro	28°41′57″ N 17°56′59″ W
Fuente Olén	28°43′47″ N 17°48′51″ W
Garafia	28°45′57″ N 17°53′59″ W
La Rosa	28°39′56″ N 17°50′55″ W
La Fajana	28°49′49″ N 17°52′11″ W
Las Caletas	28°29′44″ N 17°49′44″ W
Las Caletas II	28°42′45″ N 17°45′52″ W
Las Indias	28°30′38″ N 17°52′02″ W
Las Manchas	28°35′49″ N 17°53′20″ W
Llano Negro	28°48′15″ N 17°54′28″ W
Los Braseros	28°29′54″ N 17°49′19″ W
Los Llanos	28°39′30″ N 17°54′48″ W
LP-4 roadside	28°46′06″ N 17°54′27″ W
Malpaíises	28°34′31″ N 17°47′13″ W
Mazo	28°36′10″ N 17°46′42″ W
Mirador de Mendo	28°33′38″ N 17°51′45″ W
Mirador de los Dragos	28°45′24″ N 17°58′29″ W
Mirador del Time	28°39′48″ N 17°56′32″ W
Mirador Llano de las Ventas	28°36′48″ N 17°49′23″ W
Montaña de Mago	28°28′22″ N 17°50′19″ W
Montaña de Tagoja	28°43′16″ N 17°47′07″ W
Montes de Luna	28°31′50″ N 17°48′21″ W
Pared Vieja	28°37′10″ N 17°49′23″ W
Pico de la Cruz	28°45′04″ N 17°50′03″ W
Pino de la Virgen	28°30′35″ N 17°49′33″ W
Playa de Nogales	28°45′25″ N 17°44′09″ W
Puerto Espíindola	28°48′32″ N 17°45′47″ W
Puerto Naos	28°35′51″ N 17°53′52″ W
Puerto de Puntagorda	28°45′30″ N 18°50′16″ W
Puntagorda	28°46′11″ N 17°59′17″ W
San Andréeas	28°47′57″ N 17°45′39″ W
San Juan	28°47′05″ N 17°46′10″ W
San Juan de Puntaliana	28°44′53″ N 17°44′14″ W
Santa Cruz	28°40′27″ N 17°46′11″ W
Tazacorte	28°38′34″ N 17°56′05″ W
Tijarafe	28°42′42″ N 17°57′17″ W
Valencia	28°39′02″ N 17°50′28″ W

**Table 2 insects-14-00655-t002:** Coccinellidae recorded on La Palma. Species new to La Palma are in bold print.

Species	This Study	Literature Data
*Delphastus catalinae* (Horn)	+	[50]
*Stethorus tenerifensis* Fürsch	+	[42]
*Stethorus wollastoni* Kapur	+	[35,50,51]
*Adalia bipunctata* (L.)	+	[52]
*Adalia decempunctata* (L.)	-	[3] ^1^
*Coccinella miranda* Wollaston	+	[46,53,54,55]
*Coccinella septempunctata algerica* Kovář	+	[46,53,56]
***Harmonia quadripunctata*** (Pontoppidan)	+	-
*Hippodamia variegata* (Goeze)	+	[55]
***Myrrha octodecimguttata*** (L.)	+	-
*Olla v-nigrum* (Mulsant)	-	[57] ^2^
*Novius cardinalis* (Mulsant)	+	[35,50,54]
*Novius cruentatus* (Mulsant)	+	[34,51]
***Novius canariensis*** Korschefsky	+	-
*Clitostethus arcuatus* (Rossi)	-	[46]
***Nephaspis bicolor*** Gordon	+	-
*Nephus bipunctatus* (Kugelann)	-	[49]
*Nephus depressiusculus* (Wollaston)	-	[1] ^1^
*Nephus flavopictus* (Wollaston)	+	[46]
*Nephus incisus* (Har. Lindberg)	+	[50,58]
***Nephus reunioni*** (Fürsch)	+	-
*Scymnus cercyonides* Wollaston	+	[42,46,50,59]
*Scymnus canariensis* Wollaston	+	[46,50,51,59]
*Scymnus marinus* (Mulsant)	-	[42]
***Scymnus nubilus*** Mulsant	+	-
*Scymnus rufipennis* Wollaston	-	[46]
*Scymnus subvillosus* (Goeze)	-	[42]
*Diomus gillerforsi* Fürsch	-	[49]
*Diomus* sp.	+	-
*Cryptolaemus montrouzieri* Mulsant	+	[35,57]
*Chilocorus canariensis* Crotch	+	[46]
***Parexochomus nigripennis*** (Erichson)	+	-
*Pharoscymnus decemplagiatus* (Wollaston)	+	[34,46,50]
*Rhyzobius litura* (Fabricius)	+	[46]
*Rhyzobius lophanthae* (Blaisdell)	+	[50,60]
*Rhyzobius teresae* Eizaguirre	-	[49]

^1^ Listed on La Palma without details on collection date and locality; ^2^ Reported as *Harmonia axyridis*.

## Data Availability

Data is contained within the article.

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
