# Peer review of "The Ladybird Beetles (Coleoptera: Coccinellidae) of La Palma"

_insects, 2023, doi:10.3390/insects14070655_

Round 1
Reviewer 1 Report
All my suggestions are annotated in my revised version of the manuscript. Collecting animals in the Canary Islands needs a special permit. I would like to know if the authors did apply for such permit, in which case they should include a comment in the acknowledgements.

Author Response
Authors acknowledge careful and valuable comments of all the Reviewers. Authors accepted all corrections and most of the suggestions. Below is the detailed list of all revisions undertaken (Reviewer’s remarks are highlighted in grey)
Response to the reviewers' comments
Reviewer 1
Line 40: Why this unprecise number? – the number is changed to „55”
Lines 72-73: This is some confusing, given that Pinus canariensis is not a tree from the laurel forest. Better to say "laurel forest (or just laurisilva, given that laurisilva forest is a redundancy), Pinus canariensis forest, - changed to „laurisilva, and Pinus canariensis D. Smith forest”
Line 73: no dunes are found on this island, better use another term to refer to coastal habitats – changed to „coastal habitats”
Line 73: there is an Euphorbia scrub vegetation in the lowlands, and another very different scrub vegetation above 2000 m. The presentation of the range of habitats must be improved. If you want to included it (not necessarily) name all the important ones and ordered from low to high altitude – changed to „various kinds scrub vegetation”
Line 74: Plants of agricultural interest are much more abundant than decorative ones – changed to „In farmland areas plants of agricultural interest are abundant, and in anthropogenic habitats, decorative plants sustained by irrigation are cultivated (Figure 1).
Line 101: delete El – deleted in line 74 and in the remaining text
Line 113: Malpaíses with accent – changed to Malpaíses in line 113 and in the remaining text
Line 122: delete "of" - deleted
Line 141: Montes – changed to Montes in line 141 and in the remaining text Line 150: Malpaíses - changed to Malpaíses in line 150 and in the remaining text
Line 151: Is there on La Palma a locaton called El Pinar? –location changed to „LP-4 roadside”
Line 152: delete – the fragment marked deleted, the sentence changed to „… in North Africa.”
Line 161: Malpaíses - changed to Malpaíses Line 170: Not in bold face – changed to „El Pilar” (not in boldface)
Line 178: Breña - changed to Breña
Line 180: delete El – deleted in line 180 and in the remaining text
Line 184 (delete of) – deleted
Line 231: Montes de Luna - changed to Montes de Luna
Line 234: Malpaíses - changed to Malpaíses
Line 235: Espíndola with accent, Andrés – changed to Espíndola, Andrés
Line 243: delete "the" – deleted
Line 248: Llanos – changed to Llanos
Line 251-252: , but not from the Canary Islands.: sentence changed to „… but not from the Canary Islands.”
Line 262: Montes - changed to Montes
Line 264: delete El - deleted
Line 266: Garafía with accent, Malpaíses – changed to Garafía, Malpaíses
Line 268: Espíndola, Andrés – changed to Espíndola, Andrés
Line 277: Espíndola, Andrés – changed to Espíndola, Andrés
Line 282: unnecessary, it's obvious after the last sentence, and it is again repeated in Table 2 – Sentence „New to La Palma” deleted
Line 298: (Malpaíses) – changed to Malpaíses
Line 311: (Andrés) – changed to Andrés
Line 317: Mediterranean – changed to Mediterranean
Line 310: unnecessary: it's obvious after the last sentence, and it is again repeated in Table 2: Sentence „New to La Palma” deleted
Line: 324: Montes – changed to Montes
Line 327: Andrés - changed to Andrés
Line 338: (Montes) , Puntallana: changed to Montes, Puntallana
Line 348: (Malpaíses) – changed to Malpaíses
Line 349: (Andrés) - changed to Andrés
Line 363: delete: „specimen” deleted
Line 366: than those reported: changed to „than those reported”
Line 390-391: Not relevant sentence, given it's some speculative - sentence „The likely source of Canarian N. reunioni is either the Iberian Peninsula or Macaronesia (the Azores or Madeira).” is deleted
Line 426: Did the authors apply for a permit to collect beetles in La Palma? It is compulsery to do it, either to the Canary Government or to the Cabildo of La Palma. If this is the case, this should be stated in the aknowledgements – No, we did not apply for a permit since: 1) Coccinellidae „are not included in the Catalogue of Protected Species”, and 2) „Collecting took place outside protected areas”, then „No collecting permit needed”, according to „Application forms & guidelines” , https://www.antoniomachado.net/activities/entomology/.
Line 428: Some journal titles are abbreviated and some are not. Please, standardize them; - we standarized References by using the proper abbreviated journal names
Line 445: Coléoptères de la region: changed to Coléoptères de la région
Line 514: in italics: changed to . „Cryptolaemus montrouzieri”
Line 525: in italics: changed to „Rhyzobius”
Line 551: Commentationes Biologicae – not adopted, to standarize we used abbreviated form
Line 554-555: italics – changed to „Chamaecytisus proliferus” „palmensis”
Line 564: : Commentationes Biologicae – not adopted, to standarize we used abbreviated form
Reviewer 2 Report
The current manuscript contains faunistic data (including new records) of Coccinellidae from La Palma and will be good contribution to the knowledge of the Canarian biodiversity. It is carefully prepared and organized. Authors also summarize previously published data and provide list of Coccinellidae known from La Palma.
I haven`t any corrections and I recommend to publish this manuscript as it.
Author Response
Authors acknowledge careful and valuable comments of all the Reviewers. Authors accepted all corrections and most of the suggestions.
Reviewer 3 Report
The study presented in the manuscript “The Ladybird Beetles (Coleoptera: Coccinellidae) of La Palma” is an important contribution to a better knowledge of the Coccinellidae fauna, not only for the Canary Islands, but also for the Macaronesia Region and its putative biogeographic connection with the European and African fauna. The manuscript is, in general, well written with a clear message and appropriate tables and photos. The systematic arrangement of Coccinellidae follows a recent phylogenetics and evolutionary framework.
My major concern goes to the limited scope of the study once a single island of Canary archipelago is object of study (see 1st sentence of the abstract, lines 24-25). The absence of experimental and theoretical results makes this study short intended for the standards of a scientific paper of the category “Article”. For instance, a detailed meta-analysis using previous published data of Canary Islands together with the current study, framed in a biogeographical context, would make the study more influential. I feel that other journal aiming to publish checklists and/or database are more suitable venues for that type of biogeographic data.
Additional comments:
It is not clear the inaccuracy in the number of species recorded in the Canary Islands and La Palma (e.g., line 30 and line 40).
Authors state that ladybirds were recorded at 57 sites on La Palma (Table 1), but in table 1, only 55 sites are mentioned.
Line 92, How were individuals collected in the larval and pupae stages identified? Were the authors able to identify immature states as larvae of the genus Scymnus?
The way in which the different species were commented, especially in the "distribution" section, needs to be improved. Sometimes the authors refer to the presence of the same species in Madeira and the Azores and in others they do not.
The legends of some figures need to be improved.
Line 355, Authors mention 28 ladybird species were reported to occur on La Palma (Table 2). However, in table 2, I count 29 species. This need clarification. More, it is states that “We failed to find 10 of them…” I recommend to use the signal “-“ to indicated those species.
In the last paragraph, authors mention the expected arrival of H. axyridis in La Palma. This is a very interesting topic for discussion. This paragraph needs further developments due to its apparent establishment failures in the Azores but recent records in Madeira islands.
Not applicable.
Author Response
Authors acknowledge careful and valuable comments of all the Reviewers. Authors accepted all corrections and most of the suggestions. Below is the detailed list of all revisions undertaken (Reviewer’s remarks are highlighted in grey)
Response to the reviewers' comments
The reviewer remarks: “The absence of experimental and theoretical results makes this study short intended for the standards of a scientific paper of the category “Article”, and “I feel that other journal aiming to publish checklists and/or database are more suitable venues for that type of biogeographic data.”. We believe that by providing 1) detailed descriptions and figures, as well as taxonomic decisions on selected species, 2) a critical summary of previously published historic data, and 3) pointing attention to the rapid colonization of the island by exotic ladybird species – the paper meets the requirements of the category “Article”. We indeed plan to work on the summary on Canarian Coccinellids, framed in a biogeographical context, however, several taxonomic issues remain to be provided in separate articles earlier.
It is not clear the inaccuracy in the number of species recorded in the Canary Islands and La Palma (e.g., line 30 and line 40): the two sentences are rewritten to provide more accurate numbers:
Line 30: “…total of 35-36 species of Coccinellidae” changed to „ at least 35 species of Coccinellidae”
Line 40: „There are about 55-56 species of Coccinellidae” changed to: „There are about 55 species of Coccinellidae”
Authors state that ladybirds were recorded at 57 sites on La Palma (Table 1), but in table 1, only 55 sites are mentioned – agreed, and we found one more duplicated site (El Cubo de la Galga and Cubo de la Galga), therefor the number was changed to 54 sites: “Ladybirds were recorded at 54 sites on La Palma…” (Line 80)
Line 92, How were individuals collected in the larval and pupae stages identified? Were the authors able to identify immature states as larvae of the genus Scymnus? - we added the following sentence in Materials and Methods: „Species identification was based on morphological and anatomical features, including the form of reproductive organs (see [4, 5, 14, 16]), individuals collected in the larval and pupae stages were reared into adults in the laboratory.”
The way in which the different species were commented, especially in the "distribution" section, needs to be improved. Sometimes the authors refer to the presence of the same species in Madeira and the Azores and in others they do not – comments on the presence of the species in Madeira and Azores are given for Macaronesian ladybirds, but not in the case of species with wider geographic ranges.
The legends of some figures need to be improved – Figure 3 legend was corrected to: “Figure 3. Habitus. (A) Novius cruentatus; (B) Nephaspis bicolor; (C) Diomus sp.”
Line 355, Authors mention 28 ladybird species were reported to occur on La Palma (Table 2). However, in table 2, I count 29 species. This need clarification. More, it is states that “We failed to find 10 of them…” I recommend to use the signal “-“ to indicated those species – the information that 28 ladybird species were reported to occur on La Palma (Table 2) is correct, there is one additional record of Diomus sp. (not identified to the species) in the Table; the signal “-“ was added in Table 2 to indicate species that were not recorded in this study.
In the last paragraph, authors mention the expected arrival of H. axyridis in La Palma. This is a very interesting topic for discussion. This paragraph needs further developments due to its apparent establishment failures in the Azores but recent records in Madeira islands – we developed the paragraph by adding: “H. axyridis could not establish, despite its deliberate introductions in the Azores between 1988 and 1995 [70] and the first findings of reproducing population of H. axyridis in Macaronesia have been reported from Funchal (Madeira) in 2019 and 2020 [5, 11].” A new reference [70] was added to the References.
Reviewer 4 Report
The manuscript has added new knowledge to the Coccinellidae of La Palma. The Authors have used sound basis of their technical knowledge in the field systematic of Coccinellidae along with previous studies for managing the current update. I have some suggestions for betterment.

Author Response
Authors acknowledge careful and valuable comments of all the Reviewers. Authors accepted all corrections and most of the suggestions. Below is the detailed list of all revisions undertaken (Reviewer’s remarks are highlighted in grey)
Response to the reviewers' comments
- Add an introductory line for Coccinellidae along with updated no of species reported for this group in the world then come to Canary Islands (Line 40) - the sentence was changed to: „There are over 6000 species of Coccinellidae globally of which about 55 species were reported to occur in the Canary Islands [1,2,3,4].”
- This part is too short. You may add some review part in this section mentioning studies already done in the surrounding parts of the study region. In this paper we provide consistent review of last research on the Coccinnelids of Canary Islands, while the historic overview was given in our previous papers [4, 5, 6, 14, 15, 16] and we do not intend to repeat these review.
At least add standard methods of identification like keys followed. - we added the following sentence in Materials and Methods: „Species identification was based on morphological and anatomical features, including the form of reproductive organs (see [4, 5, 14, 16]), individuals collected in the larval and pupae stages were reared into adults in the laboratory.” You have recorded new species in this study and colored micrographs have been used. At least add which microscope system and attached camera were used. Give model for both - we added the following sentence: “Habitus images were taken using a stereo microscope Leica MZ 16 with a digital camera IC 3D attached. The photographs of the genitalia were taken using an Olympus DP23 digital camera attached to an Olympus BX43F compound microscope. Final images were produced using Helicon Focus 5.0 × 64 and Adobe Photoshop CS6 software.”
- Make this part more attractive for readers, please add distributional maps of the all recorded species from this region. Make maps using Arc Gis. – we do not consider distributional maps would provide significant information for an Island of small size. We were advised against providing maps in our earlier papers on Coccinellidae of single Canary Islands.
Round 2
Reviewer 3 Report
The authors have effectively addressed my comments and suggestions generating, by this was, an improved version of the manuscript. However, my major concerns remain unresolved, namely to the limited scope of the study once a single island of Canary archipelago is object of study. The absence of experimental and theoretical results makes this study short intended for the standards of a scientific paper of the category “Article”. For instance, a detailed meta-analysis using previous published data of Canary Islands together with the current study, framed in a biogeographical context, would make the study more influential.